# The Role of Vitamin B6 in Peripheral Neuropathy: A Systematic Review

**DOI:** 10.3390/nu15132823

**Published:** 2023-06-21

**Authors:** Raman Muhamad, Alexandra Akrivaki, Georgia Papagiannopoulou, Periklis Zavridis, Panagiotis Zis

**Affiliations:** 1Sheffield Institute for Translational Neuroscience, University of Sheffield, Sheffield S10 2HQ, UK; 2Second Department of Neurology, School of Medicine, Attikon University Hospital, National and Kapodistrian University of Athens, 12462 Athens, Greecegeorgiapap22@hotmail.com (G.P.); 3Cyprus Pain Clinic, Egkomi 2415, Cyprus; pzavridis@painclinic.com.cy; 4Medical School, University of Cyprus, Nicosia 1678, Cyprus; 5Medical School, University of Sheffield, Sheffield S10 2RX, UK

**Keywords:** nutrition, neuropathy, B6, toxicity, deficiency

## Abstract

Introduction: Vitamin B6 is a water-soluble vitamin that is naturally present in many foods and is accessible in many dietary supplements. The three natural forms are pyridoxine, pyridoxal, and pyridoxamine. Both vitamin B6 deficiency and high B6 intake have been described as risk factors for developing peripheral neuropathy (PN). The aim of this systematic review is to characterize and comprehensively describe B6-related PN. Method: A systematic, computer-based search was conducted using the PubMed database. Twenty articles were included in this review. Results: Higher vitamin B6 levels, which usually occur following the taking of nutritional supplements, may lead to the development of a predominantly, if not exclusively, sensory neuropathy of the axonal type. After pyridoxine discontinuation, such patients subjectively report improved symptoms. However, although low vitamin B6 levels can be seen in patients suffering from peripheral neuropathy of various etiologies, there is no firm evidence that low B6 levels have a direct causal relationship with PN. Many studies suggest subjective improvement of neuropathy symptoms in patients suffering from PN of various etiologies after receiving B6 supplementation; however, no data about B6 administration as a monotherapy exist, only as part of a combination treatment, usually with other vitamins. Therefore, the potential therapeutic role of B6 cannot be confirmed to date. Supplementation with vitamin B6, even as part of a nutritional multivitamin supplement, has not been proven harmful at permitted daily doses in patients who already suffer from PN. Conclusion: Current scientific evidence supports a neurotoxic role of B6 at high levels. Although some studies suggest that low B6 is also a potential risk factor, further studies in this area are needed.

## 1. Introduction

Polyneuropathy (PN) is a common neurological disorder that affects the peripheral nervous system. It encompasses a wide spectrum of clinical syndromes, depending on the anatomic regions of the peripheral nervous system that are affected. Patients may present with symptoms such as paresthesia, hypesthesia, weakness, atrophies, reduced or diminished reflexes, and pain [1]. 

The prevalence of PN is estimated to exceed 5% of the general population older than 50 years of age [2], with higher rates among older adults and those with underlying medical conditions [3]. Causes of PN include genetic factors, infections, metabolic disorders, exposure to toxins, and systematic diseases, although in some cases, its etiology remains idiopathic despite extensive investigations [4]. The role of nutrition in the pathogenesis of PN has been studied, and the crucial role of specific vitamins, such as B12, has been confirmed [5]. 

Vitamin B6 is a water-soluble vitamin that is naturally present in many foods and is accessible in many dietary supplements. The three natural forms are pyridoxine, pyridoxal, and pyridoxamine. The biologically active form is pyridoxal-phosphate (PLP), which is a crucial coenzyme in numerous enzymatic activities related to the metabolism of carbohydrates, proteins, and lipids, as well as the production of neurotransmitters [6]. The recommended daily intake of dietary vitamin B6 to maintain acceptable nutritional status is between 1.6 and 2 mg/day [7]. Pyridoxine, like cyanocobalamin (B12) and folate (B9), is important for the conversion of methionine into cysteine and is hence essential for nerve function and the survival of neurons [8]. 

Insufficient intake or poor absorption of vitamin B6 can result in B6 deficiency. Particularly at risk are certain populations with conditions that increase metabolic needs. Additionally, isoniazid, phenelzine, hydralazine, penicillamine, levodopa, and chemotherapy treatments are a few medications that might lead to pyridoxine shortages because they interfere with its metabolism. In contrast, excessive intake of vitamin B6 supplements is a risk factor for toxicity [9].

Pyridoxine deficiency has been linked to several neurological and non-neurological conditions, such as anemia, dermatitis, glossitis, depression, confusion, a weakened immune system, and PN [10]. In contrast, the association of high intake of pyridoxine with PN has been presented in several cases in the absence of other potential etiologies [11]. The exact pathophysiological mechanisms through which B6 is linked to neuropathy are not known.

The aim of this systematic review is to characterize and comprehensively describe the role of vitamin B6 in PN.

## 2. Materials and Methods

### 2.1. Literature Search Strategy 

A systematic MEDLINE search for all published papers referring to B6 and PN was performed using PubMed. We included all eligible included papers up to 1 June 2022. For the search, two Medical Subject Heading (MeSH) terms were used. Term A was “neuronopathy” or “ganglionopathy” or “neuropathy” or “polyneuropathy”; and term B was “B6” or “pyridine” or “pyridoxine” or “pyridoxal” or “pyridoxamine”. An English language filter was applied in the search. 

All study data were aggregated and referenced in accordance with the PRISMA (Preferred Reporting Items for Systematic Reviews and Meta-analysis) guidelines.

### 2.2. Inclusion and Exclusion Criteria

All articles were reviewed for inclusion by two authors. When there was uncertainty or conflict in decisions, it was discussed to reach a consensus. Articles eligible to be included in this review were required to meet the following criteria:The study was conducted using human subjects;The article was written in the English language;The study discussed possible links between B6 and PN.

Articles meeting the following criteria were excluded from the systematic review:Non original studies (i.e., reviews/letters to the editor/opinion papers);Case reports or case series with fewer than five participants with B6-related PN;Papers not related to PN;Papers not related to B6;Full texts not in English;Low-quality trials measured as by Jadad score, where applicable.

### 2.3. Data Collection Process

Data were extracted from each study according to pre-agreed study outcomes. Data collected included: study type (observational/interventional), B6 status (deficiency, excess, given prophylactically, given to treat), number of participants, number of participants with PN, clinical type of PN (sensory, motor, sensory motor), neurophysiological type of PN (axonal, demyelinating), type of assessment tool used for PN (questionnaires, nerve conduction studies (NCS), quantitative sensory testing (QST), and biopsies), and main outcome of the study. 

### 2.4. Risk of Bias in Individual Studies

The Jadad score was used to examine the bias in clinical trials. The Jadad scale classifies the quality of randomized, controlled trials and includes only five items depending on the study randomization: blindness of participants and investigators, blindness in outcome assessments, reports of withdrawals, and dropouts. Each item, if present, receives 1 point. Trials with a Jadad score less than 4 were considered to have a significant risk of bias and were excluded [12]. Since Jadad scoring only considers risk of bias, additional factors impacting trial quality, such as the population size, power, and length of follow-up, are addressed in the text.

### 2.5. Synthesis of Results

Due to the diversity of the gathered literature, this study offers its data in a narrative format. There were no measures of consistency in place to report on the results of included papers.

### 2.6. Compliance with Ethical Guidelines

Considering that only previously published papers were used for the systematic review, there are no ethical concerns with this study.

### 2.7. Study Registration

This review was conducted for the purposes of the dissertation of the first author during his MSc studies in clinical neurology at the University of Sheffield under the registration ID MEDT21.

## 3. Results

### 3.1. Nature of Included Studies

The above-described literature search strategy revealed 219 papers. These papers underwent an abstract screening process, and 186 articles were excluded; 33 articles remained for full text eligibility screening. During this process, 13 articles were further excluded, and a total of 20 articles remained and were included in the review [13,14,15,16,17,18,19,20,21,22,23,24,25,26,27,28,29,30,31,32]. Figure 1 illustrates the study selection process (PRISMA chart), and Table 1 provides a brief summary of the included studies. The reasons for exclusion for the excluded papers as well as the quality assessment of the included RCTs are available as Appendix A.

### 3.2. Neuropathy due to B6 Deficiency

Six studies attempted to describe the role of B6 deficiency in the development of PN [16,19,20,26,29,31].

Loens et al., in their retrospective, observational study of patients with advanced idiopathic Parkinson’s disease (IPD), sought interactions between levodopa medication and plasma levels of B vitamins and to investigate the prevalence of PN and its relationship with B vitamin plasma levels [16]. Participants were given either oral levodopa therapy (*n* = 13) or levodopa/carbidopa intestinal gel (LCIG) (*n* = 8). Fourteen of 17 patients (81%) with available electrophysiological data had PN, and in most cases, this disease was axonal in nature. Levodopa is a major risk factor for PN in a dose-dependent manner; this relationship is supported by the authors’ discovery that patients with LCIG exhibited more severe axonal damage and more nerve injuries than the oral group. Patients receiving LCIG had a daily levodopa dose (LDD) that was more than 2.5 times higher than in the oral levodopa therapy group. Pyridoxine was also significantly lower in the LCIG group, leading the authors to hypothesize that carbidopa’s effects in LCIG could be the cause. The decarboxylase inhibitor known as carbidopa is commonly used in LCIG and in most oral levodopa formulations. Carbidopa depletes the pyridoxine reserve pool by irreversibly binding to and permanently deactivating free pyridoxine, as well as pyridoxine-dependent enzymes, such as various decarboxylases. However, this study has some significant limitations, including the retrospective nature of the study, the failure to exclude other potential risk factors that can lead to PN, and the small sample size. 

Similarly, Pauls et al. reported that, of a cohort of 19 patients on LCIG (of which six were on B-vitamin substitution), two patients (11%) developed new-onset PN after initiation of LCIG therapy (none of whom were on vitamin substitution) [31]. They concluded that vitamin B substitution appears to reduce coupling between levodopa dosage and homocysteine and may be useful to prevent polyneuropathy related to LCIG.

Moriwaki et al., in their prospective, observational study of patients on chronic peritoneal dialysis (*n* = 66), aimed to determine any correlation between the level of vitamin B6 and symptoms of PN [26]. Patients were asked to respond to a questionnaire that focused mainly on two symptoms: burning and painful paresthesia. Twelve of the 66 patients in the study reported having at least one sensory problem. The serum levels of PLP obtained from the patients who complained of sensory impairment were significantly lower than those obtained from patients who had no such complaints, suggesting a correlation between B6 and neuropathic symptoms. To explain this correlation, several causes for altered vitamin B6 metabolism related to uremia were suggested, such as inadequate intake or absorption, dialysis loss, suppression of PLP action or metabolism by uremia toxins, impaired pyridoxal kinase-mediated phosphorylation of pyridoxal, and enhanced PLP phosphatase degradative activity. A significant limitation of this study, however, was that uremia is a known risk factor of PN; therefore, at least some of the signs and symptoms of PN described can be attributed to uremia.

McCann and Davis, in their case-controlled study [29], measured serum vitamin B6 levels in 50 patients with diabetic PN to investigate whether there is any correlation between diabetic PN and serum B6 levels. Pyridoxal deficiency was present in 25% of the patients irrespective of the duration of diabetes or the treatment employed. The control group consisted of age- and gender-matched diabetic patients without PN. Patients with PN had a significantly lower mean serum pyridoxal concentration compared to controls. The authors debated whether the lack of a relationship among the duration, severity of diabetes, and diabetic PN in this cohort suggested that low serum pyridoxal concentrations may be an additional risk factor contributing to the development of PN. A hypothesis of the authors was that low pyridoxal levels was linked to poor control of diabetes. It has been shown that pyridoxal deficiency increases the concentrations of the tryptophan metabolite xanthurenic acid, which binds to insulin, and that this complex has a significantly lower ability to lower blood glucose levels than native insulin. The authors concluded that pyridoxal (B6) deficiency must be considered one of the potential metabolic factors that may cause PN in patients with diabetes mellitus and that measurements of serum pyridoxal should be routinely requested in diabetic patients suffering from PN.

van der Watt et al., in their prospective, observational study [19] of human immunodeficiency virus (HIV)-infected individuals (*n* = 159) before or after antiretroviral therapy (ART), investigated the connection among history of tuberculosis (TB), pyridoxine deficiency, slow acetylation phenotype, and PN. Fifty-three percent of the cohort had pyridoxine deficiency. PN was present in 26 patients (16%). After initiating ART, 25 (19%) patients developed PN within 24 weeks, and seven patients who already had PN at baseline had worsening of symptoms. However, the authors found no correlation between pyridoxine deficiency and PN. What limited the credibility of these results is that 97% of the cohort were receiving a low dose of pyridoxine supplementation during the study. 

Centner et al., with their prospective, observational study [20], aimed to determine the relationship of plasma levels of pyridoxal 5 phosphate (PLP), the active coenzyme form and a marker of total body vitamin B6, and plasma levels of 4-pyridoxic acid (4PA) with sensory PN in HIV-infected patients receiving TB therapy (*n* = 116). Sixty-five patients (56%) had PN at the beginning of the trial. Patients with PN did not exhibit significant differences in PLP or 4PA at any sampling point compared to participants without PN. Further to this finding, patients developed PN despite receiving pyridoxine supplements and having normal plasma PLP levels, a finding suggesting that other mechanisms are likely the cause of PN in these patients. 

### 3.3. Neuropathy due to B6 Toxicity

Six studies attempted to describe the potential role of B6 excess in the development of PN [14,17,18,22,24,28].

Latov et al., in their retrospective, observational study of a cohort of 137 patients with PN, identified 23 patients with PN that had no other causes for the development of PN than their nutritional status [18]. All 23 patients had pure sensory PN, 14 of whom had large fiber axonal PN, and 9 had pure small fiber neuropathy. Ten of these 23 patients (five with large fiber and five with small fiber neuropathy) had elevated PLP levels as their only abnormality.

Court et al. studied prospectively over 12 weeks 144 patients with multidrug-resistant tuberculosis treated with terizidone to determine risk factors for PN [14]. Since cycloserine and its structural analogue terizidone have been linked to PN, high-dose pyridoxine (150 or 200 mg daily) was administered as prophylaxis for terizidone-related pyridoxine deficiency. During the study, 50 (35%) patients were diagnosed with PN. Although pyridoxine was given for prophylaxis for terizidone-related PN, the results showed that patients prescribed the 200-mg daily dose had a 2.8 times greater risk of PN compared to those on 150 mg of pyridoxine daily (*p* = 0.012). An additional interesting finding was that the median time to PN in participants receiving 200 mg of pyridoxine daily was shorter in comparison with participants receiving 150 mg of pyridoxine daily (38.5 days compared to 43 days, respectively). 

Alsabah et al., in their case-controlled study, examined factors associated with the development of PN in 32 patients who underwent laparoscopic sleeve gastrectomy (LSG) [17]. The follow-up period was up to 18 months after surgery. All patients received supplementation with multivitamin tablets that included vitamin B1 (100 mgs daily), vitamin B6 (200 mgs daily), and vitamin B12 (200 μg daily). Of the 32 patients enrolled, 16 developed PN. B6 levels in the PN group were higher than normal, which was unsurprising considering the high dose of B6 supplementation of 200 mg a day, which is much higher than the upper limit of 100 mg/day set by the FDA, whereas those in the control group were within normal limits. Vitamin B12 levels in the study were within the normal range in both the PN and no-PN groups. However, thiamine levels were significantly reduced in the PN group, which is also a known risk factor for the development of PN. 

Scott et al., in their retrospective, observational study, sought a possible relationship between elevated B6 and PN [24]. They identified 26 patients with elevated B6 and PN. All patients had sensory PN. Nine had large fiber axonal PN, whereas 17 were considered to suffer from small fiber PN. Half of the patients reported pain. A significant limitation of this study included the lack of information about the duration of the B6 excess. Moreover, although patients were advised to stop receiving B6 supplements, and the majority reported that their symptoms stabilized, a detailed clinico-electrophysiological follow-up did not occur.

Parry and Bredesen, in their observational study performed in a group of 16 patients with symmetrical sensory PN associated with pyridoxine abuse for up to 72 months [28], found that electrophysiology reveled axonal sensory loss. A biopsy of the sural nerve that was performed on two patients showed moderate reduction in the density of myelinated fibers and axonal degeneration. No segmental demyelination or myelinated fiber regeneration was demonstrated. After pyridoxine discontinuation, patients had improved signs or symptoms, but none had their symptoms or signs completely resolved. 

Visser et al. conducted a prospective, case-control study to determine the potential role of B6 in the pathogenesis of PN [22]. In this study, B6 supplementation and serum B6 levels were calculated in a group of 381 patients with CIAP and 150 controls without PN. Although more patients (31%) than controls (22%) used supplements containing vitamin B6, the levels of vitamin B6 in patients were not notably higher than in the controls. The severity of PN did not significantly correlate with vitamin B6 levels. Follow-up of patients confirming the cessation of supplements showed slow progression of symptoms in 64%, stabilization in 26%, and regression in 10%. 

### 3.4. B6 Used for the Treatment of Neuropathy

In eight studies, B6 supplementation was studied for the management of PN [13,15,21,23,25,26,27,30].

Moriwaki et al., during their study [26], administered B6 in a dose of 30-mg pyridoxine hydrochloride, along with 15 mg of thiamine, 10 mg of riboflavin, 50 mg of niacin, and 10 mg of pantothenic acid. Following four weeks of supplementation with vitamin B6, the plasma levels of PLP in all patients with symptoms matched those in patients without complaints, and this rise was accompanied by an improvement in the signs and symptoms in eight out 12 patients, further suggesting a link between B6 and PN. 

Stewart et al. in their cross-sectional study investigated 261 patients with chronic idiopathic axonal PN (CIAP) to see how vitamin B6 levels correlated with the severity of PN [13]. Borderline elevated B6 plasma levels (50–100 μg/L) were seen in 15.9% of the patients, while 16.3% had a B6 level greater than 100 μg/L. As expected, levels of vitamin B6 were much greater in those using certain supplements (multivitamins, B6 supplements, and B complex vitamins), but their symptoms were not correspondingly worse. There was no correlation among an elevated vitamin B6 level, signs, and clinical severity. Overall, the results suggest that supplementation with vitamin B6 has no discernible impact on the symptoms of PN in people who have already experienced PN. Even in the group of patients with the increased plasma level of B6, PN signs or symptoms were not worse. One major limitation was this study’s use of cross-sectional data, which prevented the authors from identifying changes in patients’ symptoms before, during, and after vitamin supplementation. Another limitation was that no data were available for duration or dose of vitamin B6 supplementation.

Aydin Köker et al., in their prospective, observational study, examined the use of pyridoxine (150 mg/m^2^ BID) and pyridostigmine (3 mg/kg BID) in the treatment of vincristine-induced PN in 23 pediatric patients with acute lymphoblastic leukemia [15]. The results showed that, when using pyridoxine combined with pyridostigmine therapy, symptoms of PN improved dramatically. However, significant limitations of this study included the small sample size of patients and the absence of a control group. Another limitation is we could not examine the benefit of pyridoxine therapy alone since it was given in combination with pyridostigmine. Additionally, no serum pyridoxine levels were available before or after the initiation of treatment. This information would have helped to show whether a deficiency of B6 or a direct effect of B6 treatment truly had effects on PN.

Trippe et al., in their prospective, observational study, evaluated changes in symptoms and quality of life after administering for 12 weeks a combination of L-methyl folate, methylcobalamin, and pyridoxal-phosphate (LMF-MC-PLP) in 544 patients with diabetic PN [21]. Neuropathy symptoms, including pain, significantly improved between baseline and follow-up (*p* < 0.001). Moreover, LMF-MC-PLP led to a significant reduction in the amount of symptom-related disruption with patients’ regular daily activities (*p* < 0.001). However, limitations of this study included the lack of a control group and the use of subjective measures of patients’ symptoms.

Fonseca et al. conducted a randomized, double-blind, placebo-controlled trial involving patients with diabetic PN (*n* = 214) who were randomly assigned to 24 weeks of treatment with the LMF-MC-PLP combination versus placebo [23]. The primary outcome, which was the impact on VPT (vibration perception threshold), which was measured by a VPT meter on the great toe of each foot, did not substantially differ between the LMF-MC-PLP and placebo groups over the entire duration of the study. However, regarding secondary outcomes, patients receiving LMF-MC-PLP reported clinically significant improvements in neuropathy symptoms at weeks 16 and 24 compared to placebo. Improvement in symptom scores was inversely associated with baseline levels of PLP (*p* = 0.003) and positively associated with changes in PLP (*p* = 0.003), suggesting that symptom improvement was probably influenced by the PLP component in the combination of LMF-MC-PLP.

In another randomized, double-blind, placebo-controlled trial, Peters et al. assessed over a period of 12 weeks the therapeutic effectiveness of taking a supplement containing B1 (250 mg), vitamin B2 (10 mg), vitamin B6 (250 mg), and vitamin B12 (0.02 mg) three times daily versus a supplement containing the same vitamins plus vitamin B9 (1 mg) versus placebo in 325 patients suffering from alcohol-related PN [25]. The results showed that, in comparison to placebo, patients receiving either nutritional supplement demonstrated a substantial improvement in both the primary efficacy endpoint (vibration perception threshold at the big toe) and secondary efficacy endpoints, which included pain. However, major limitations of this study included the lack of measurements of B6 levels before and during the study and that the positive effect could not be clearly attributed to a specific nutrient in the combination that was administered.

Okada et al., in their prospective, case-control study of 26 patients with chronic renal failure on high flux hemodialysis (HD) and human recombinant erythropoietin suffering from PN, compared B6 supplementation at a dose of 60 mg/day (*n* = 14) versus B12 supplementation at a dose of 500 μg/day (*n* = 12) [27]. Although vitamin B6 deficiency was not demonstrated through biochemical tests in patients with chronic renal failure on high-flux HD, B6 supplementation was effective in improving PN symptoms, whereas no improvement was observed in response to vitamin B12 supplementation in the control group. However, limitations of this study included the small sample size, the unmatched control and treatment groups, and the short duration of follow-up (4 weeks).

Devadatta et al., in their prospective, interventional study, examined the effectiveness of B6 (administered alone or as part of a vitamin B complex supplement) in 16 poorly nourished tuberculous patients receiving isoniazid and who had developed PN [30]. Patients were treated differently since the dose of B6 that was administered varied, and some discontinued isoniazid, while others did not. However, all patients reported improvement, even those who did continue isoniazid treatment and received B6 alone for the management of their symptoms. The non-randomized nature, the small sample size, and the neurotoxic effect of isoniazid were major limitations of this study.

### 3.5. Genetic Predisposition

Chelban et al. described the presence of biallelic mutations in PDXK in five individuals from two unrelated families with primary axonal PN and optic atrophy [32]. The natural history of this disorder suggests that untreated, affected individuals become wheelchair-bound and blind. The authors identified conformational rearrangement in the mutant enzyme around the ATP-binding pocket. Low PDXK ATP binding resulted in decreased erythrocyte PDXK activity and low PLP concentrations. PLP supplementation led to a favorable outcome.

## 4. Conclusions

In summary, the present study draws the following conclusions regarding B6 and PN.

Low vitamin B6 levels can be seen in patients suffering from peripheral neuropathy of various etiologies. However, this finding is due either to overall poor nutritional status (which means low levels of other vitamins, deficiencies in which can cause PN, such as B12) or to an adverse effect of treatments that these patients received for conditions that can lead to neuropathy, such as diabetes or chronic renal failure. Therefore, to date, there is no firm evidence that low B6 levels have a direct causal relationship with PN. Measuring levels of B6 in patients with CIAP who receive no nutritional supplements could shed light on this association and should be undertaken as a research project in the future.

Higher vitamin B6 levels, which usually occur following the taking of nutritional supplements, may lead to the development of a predominantly, if not exclusively, sensory neuropathy of the axonal type. After pyridoxine discontinuation, such patients do subjectively report improved symptoms. However, the studies available to date have had small sample sizes, and the evaluation of patients has mainly included the presence of symptoms and not detailed clinical or electrophysiological evaluations. 

Many studies have suggested a subjective improvement of neuropathy symptoms in patients suffering from PN of various etiologies after receiving B6 supplementation. In none of those studies, however, has B6 been administered as a monotherapy but as part of a combination treatment, usually with other vitamins. Therefore, the potential therapeutic role of B6 cannot be confirmed to date. 

In contrast, supplementation of B6 vitamins, even as part of a nutritional multivitamin supplement, has not proven to be harmful at permitted daily doses in patients who already suffer from PN.

## 5. Limitations

We excluded case reports and series of fewer than five cases. Such studies provide a low quality of evidence by definition, which was why we chose not to include them in our review. However, there is a small possibility that we might have missed some information that could have been of interest.

In the included studies, patients’ dietary intake differences that could have confounded the effects of B6 loading (i.e., both high and low) were not considered. This fact may be a source of bias, as it is in all studies of nutrients that can be absorbed through the diet. 

Similarly, the included studies were performed in different populations; therefore, regional differences in populations that may be subject to epigenetic modifications have not been considered. 

Not all studies managed to provide data about the time of B6 deficiency or B6 excess before the onset of neuropathy. Such information would have been very useful in understanding B6-related peripheral neuropathy and should be included in future studies in the field.

There were no restrictions on the date of publication applied in this review. This decision was deliberate to assure the review of the full range of literature pertinent to the topic in question. However, since the included literature dates as far back as 1960, it is feasible that modern investigations may have identified causes of neuropathy other than B6 among some included patients. 

## Figures and Tables

**Figure 1 nutrients-15-02823-f001:**
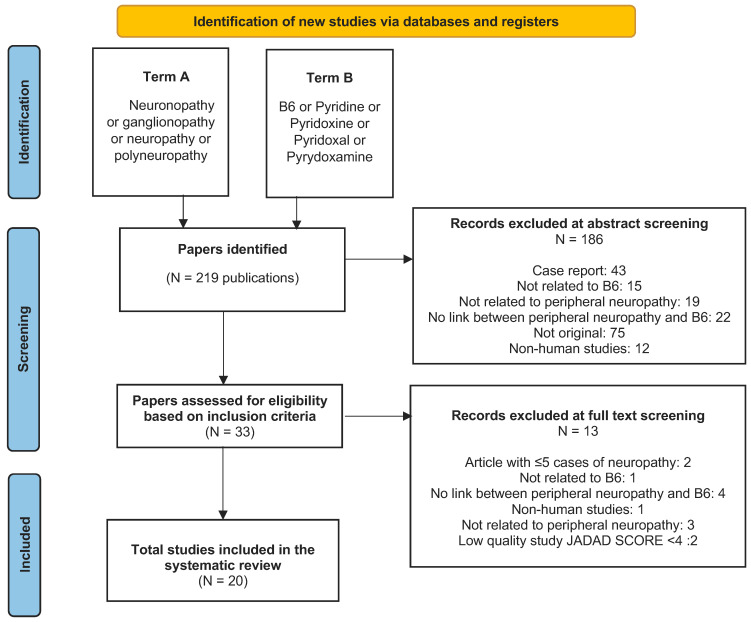
A PRISMA chart illustrating the reasons for exclusion.

**Table 1 nutrients-15-02823-t001:** A summary of the findings of included studies.

Study (Reference)	Study Type	Participants	Country	B6 Link	Type of Neuropathy	Assessment Tool for Neuropathy	Main Outcome
1-(Stewart et al., 2022) [13]	Cross-sectional study	Patients with chronic idiopathic axonal polyneuropathy (CIAP), *n* = 261 (56% males)	USA	Management of PN	Sensory motor neuropathy	Total Neuropathy Score-reduced (TNSr) + Nerve Conduction Studies (NCS)	Mild to moderate elevation of B6 (100 to 200 μg/L range) was not related to worse PN symptoms in chronic idiopathic axonal polyneuropathy (CIAP)
2-(Court et al., 2021) [14]	Prospective, observational study	Patients with tuberculosis treated with terizidone receiving prophylactic supplementation with B6 (150 or 200 mg daily), *n* = 144 (60% male)	South Africa	Excess B6	Sensory neuropathy	The Brief PN Rating Screen (BPNS)	Patients prescribed higher B6 doses had 2.78 times the risk of PN compared to those on lower dose (*p* = 0.012)
3-(Aydin Köker et al., 2021) [15]	Prospective, observational study	Pediatric patients with VIPN (vincristine-induced peripheral neuropathy) receiving prophylactic supplementation with pyridoxine 150 mg/m^2^ BID and pyridostigmine 3 mg/kg BID, *n* = 23 (44% male)	Turkey	Management of PN	Sensory motor neuropathy	World Health Organization (WHO) neurotoxicity score (sensory/motor) and the National Cancer Institute Common Terminology Criteria for Adverse Events (NCI CTCAE) + NCS	Statistically significant improvement in PN symptoms with B6 supplementation
4-(Loens et al., 2017) [16]	Retrospective, observational study	Patients with advanced idiopathic Parkinson’s disease (IPD) receiving oral levodopa therapy or levodopa/carbidopa intestinal gel (LCIG), *n* = 21 (52% male)	Germany	Low B6	Axonal sensory or sensory motor neuropathy	NCS	Lower levels of B6 due to levodopa therapy were associated with presence of more severe PN
5-(Alsabah et al., 2016) [17]	Retrospective, case-control study	Post-LSG (laparoscopic sleeve gastrectomy) patients receiving B6 supplementation 202 mg daily, *n* = 32 (16% male)	Kuwait	Excess B6	Not specified	Assessment was mainly based on symptoms of PN	Higher levels of B6 were associated with presence of PN
6-(Latov et al., 2016) [18]	Retrospective, observational study	Patients with recently diagnosed PN caused by pathological nutritional status, *n* = 23	USA	Excess B6	Sensory small fiber neuropathy (SFN)	Signs and symptoms of PN + NCS and biopsy	High pyridoxal phosphate (PYP) levels were identified in patients with neuropathy; 50% were diagnosed with SFN
7-(van der Watt et al., 2015) [19]	Prospective, observational study	HIV-infected individuals before or after antiretroviral therapy (ART), *n* = 159 (31% male)	South Africa	Low B6	Distal sensory neuropathy (DSP)	The Brief Peripheral Neuropathy Screen (BPNS) and the modified version of the Total Neuropathy Score (TNSr)	No association between pyridoxine deficiency and overall DSP
8-(Centner et al., 2014) [20]	Prospective, observational study	HIV-infected patients receiving TB therapy and prophylactic B6 supplementation (25 mg daily *n* = 88, 50–150 mg daily *n* = 28), *n* = 116 (45% male)	South Africa	Low B6	Sensory neuropathy	The Brief Peripheral Neuropathy Screen (BPNS);Neuropathic symptoms of pain, paresthesia or numbness were quantified using a visual numerical rating scale (NRS)	HIV-infected patients receiving TB treatment developed sensory polyneuropathy (SPN) at a high rate, despite receiving pyridoxine and having normal plasma B6. The SPN group did not show significant changes in plasma B6 compared with the SPN-free group
9-(Trippe et al., 2016) [21]	Prospective, observational study	Patients with diabetic PN receiving B6 supplementation (35 mg daily), *n* = 544 (46% male)	USA	Management of PN	Sensory neuropathy	Neuropathy Total Symptom Score-6 (NTSS-6) questionnaire	Improvement of PN symptoms with B6 supplementation
10-(Visser et al., 2014) [22]	Prospective, case-control study	Patients with CIAP (chronic idiopathic axonal polyneuropathy) *n* = 381 (70% males)	Netherlands	Excess B6	Sensory motor neuropathy	The sensory sum score, which ranges from 0 to 28, and a Medical Research Council (MRC) scale, resulting in a motor sum score from 0 to 40 for both lower limbs + NCS	No association between CIAP and elevated vitamin B6 serum levels
11-(Fonseca et al., 2013) [23]	Randomized, double-blind, placebo-controlled trial (Jadad score 4)	Patients with type 2 diabetes and PN receiving B6 supplementation (35 mg daily) or placebo, *n* = 214 (69% male)	USA	Management of PN	Sensory neuropathy	Vibration perception threshold (VPT) and Neuropathy Total Symptom Score-6 (NTSS-6) and others (NDS, SF-36)	VPT did not differ significantly between B6 supplementation and placebo groups; NTSS-6 scores improved significantly in patients receiving supplementation
12-(Scott, Zeris and Kothari, 2008) [24]	Retrospective, observational study	Patients with PN and elevated B6 levels, *n* = 26	USA	Excess B6	Sensory small fiber neuropathy (SFN)	Signs and symptoms of peripheral neuropathy + NCS and Quantitative Sensory Testing (QST)	Elevated B6 levels should be considered in the differential diagnosis of any sensory or sensorimotor PN
13-(Peters et al., 2006) [25]	Randomized, double-blind, placebo-controlled trial (Jadad score 4)	Patients with alcoholic PN receiving B6 supplementation (250 mg TID) or placebo, *n* = 325 (75% male)	Poland and Ukraine	Management of PN	Sensory neuropathy	Vibration perception threshold (VPT), McGill’s pain questionnaire and 2-point discrimination test	Improvement of PN symptoms after B6 supplementation
14-(Moriwaki et al., 2000) [26]	Prospective, observational study	Patients on chronic peritoneal dialysis with PN symptoms, *n* = 12	Japan	Low B6 Management of PN	Sensory neuropathy	Symptoms were assessed, focusing specifically on burning and painful paresthesia	Lower B6 levels were associated with PN symptoms; supplementation improved sensory abnormalities in 8 of 12 patients
15-(Okada et al., 2000) [27]	Prospective, case-control study	Patients with chronic renal failure on high flux hemodialysis (HD) suffering from PN receiving B6 supplementation (60 mg daily), *n* = 14 (43% male) versus B12 supplementation (500 μg daily), *n* = 12 (50% male)	Japan	Management of PN	Sensory neuropathy	Peripheral polyneuropathy (PPN) was assessed using a score of 0–4, ranging from no symptoms to painful symptoms	Improvement of PN symptoms after B6 supplementation
16-(Parry and Bredesen, 1985) [28]	Observational study	Patients with PN receiving B6 supplementation (0.2–5 g daily), *n* = 16 (0% male)	USA	Excess B6	Sensory neuropathy	Signs and symptoms of peripheral neuropathy + NCS and biopsy	B6 excess causes pure sensory, length-dependent, axonal neuropathy; improvement followed discontinuation of B6.
17-(McCann and Davis, 1978) [29]	Prospective, case-control study	Patients with diabetic PN and controls without PN, *n* = 50 (48% males)	Australia	Low B6	Sensory motor neuropathy	Signs and symptoms of peripheral neuropathy were assessed	Lower B6 concentrations in patients with PN compared with diabetic patients without PN.
18-(Devadatta et al., 1960) [30]	Prospective, interventional study	Poorly nourished tuberculous patients under isoniazid treatment with PN receiving B6 supplementation (200 mg or 6 mg), *n* = 16	India	Management of PN	Sensory motor neuropathy	Signs and symptoms of peripheral neuropathy were assessed	Improvement of neuropathy symptoms after B6 supplementation
19-(Pauls et al., 2021) [31]	Retrospective, observational study	Patients with advanced idiopathic Parkinson’s disease (*n* = 19) receiving levodopa/carbidopa intestinal gel (LCIG), *n* = 19 (47% male)	Finland	Low B6	Sensory neuropathy	Signs and symptoms of peripheral neuropathy + NCS	B6 supplementation (3 mg of B6) reduced the odds of developing PN
20-(Chelban et al., 2019) [32]	Prospective, observational study	5 individuals (1 male) from 2 unrelated families with primary axonal polyneuropathy and optic atrophy and biallelic mutations in PDXK	Cyprus, Scotland, and Italy	Low B6, management of PN	Sensory motor neuropathy	Signs and symptoms of peripheral neuropathy + NCS	Improvement of neuropathy symptoms after B6 supplementation (50 mg of B6)

## Data Availability

The data presented in this study are available on request from the corresponding author.

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
