# Peer review of "The Role of Vitamin B6 in Peripheral Neuropathy: A Systematic Review"

_nutrients, 2023, doi:10.3390/nu15132823_

Round 1
Reviewer 1 Report
In this manuscript the authors investigate the role of vitamin B6 in peripheral neuropathy as a systematic review. They analyzed a total of 18 articles (following application of inclusion and exclusion criteria). They concluded that current scientific evidence supports a neurotoxic role of B6 at high levels and also, although some studies suggest that low B6 is also associated with polyneuropathy, further studies are needed to draw a firm conclusion.
This study is performed and presented according to academic rules of scientific writing, the results are novel and relevant and provide an advance in the field.
I have only 2 minor suggestions for the authors:
- omit numbers in lines 81-86
- explain Jadad scoring system in more details
Without further a do, I would reccomend this manuscript for publication.
Author Response
Thank you for your very positive feedback.
We have made both recommended changes
Reviewer 2 Report
The artcle represents an attempt to clarify the role of vitamin B6 in peripheral neuropathy. Indeed, the aim was described as a systematic review to characterize and comprehensively describe B6 related peripheral neuropathy, but information about the search and 'data collection process' seem to dominate. The article offers very little and no new/additional information about the toxicity of B6.
The chart as well as the table are unattractive and space consuming.
The Engkish writing is O.K.
.
Author Response
Thank you for your comments.
As this is a systematic review we had to be very clear about the way this was conducted and how we ended up with including 20 papers in our review. We think that this is important to be kept in, but if also the editor wants us to reduce the information or add it a Supplement we are more than happy to do it. From our experience in all systematic reviews such information should be in the main text.
The chart (PRISMA chart) is also a very important graph for the review and has to be included as per PRISMA guidelines. Unless by "unattractive" you mean the stylistic details (i.e. colour) which again we are happy to change if needed.
Following the comments of the Editor and the other two reviewers we made some changes to the Table which is a summary of these 20 papers. If the editor prefers it we can add this Table as Supplementary material rather than in the main text.
Finally, regarding your comment about providing new / additional information: as this is a review we did not aim to provide new information but only to describe what is available in the literature and sum this information up.
Reviewer 3 Report
Instead of using the phrase "On the other hand," when starting a sentence...use "Alternatively," instead.
The manuscript does not address patient's dietary intake differences that could confound the effects of B6 loading (i.e., both high or low) and perhaps regional differences in populations that may be subject to epigenetic modifications from contaminants/neurotoxicants within their water supply. This is an overlooked critical factor.
Further, what happens when someone takes a Super B Complex vitamin? This is not addressed in the manuscript. Another crucial factor is when the patient is diagnosed with PN, is it at that time-point of diagnosis that they are determined to have altered B6 levels, or did the B6 deficiency or excess occur prior to diagnosis. This is not clear.
Table 1 should include another 2 columns with sex of the patients and ethnicity/region they were from. Additionally, a summary chart/bar graph of the demographics obtained should be presented to help visually summarize the key/main point of the manuscript.
Additionally, the references seem too little for such a review in which 70-80 papers would be more appropriate.
None.
Author Response
We would like to thank the reviewer for the very helpful points.
We have made some changes in our manuscript according to the reviewer's suggestions. Please find below our point by point replies:
- Instead of using the phrase "On the other hand," when starting a sentence...use "Alternatively," instead
We have removed the phrase "on the other hand". We did not use the word "alternatively" as the meaning of the sentences were we used "on the other hand" would be different - but we used other expressions as needed.
2. The manuscript does not address patient's dietary intake differences that could confound the effects of B6 loading (i.e., both high or low) and perhaps regional differences in populations that may be subject to epigenetic modifications from contaminants/neurotoxicants within their water supply. This is an overlooked critical factor.
This is a valid point. However, the majority of the papers that met the inclusion criteria to be included in the review did measure the serum B6 levels which strengthened their results. But as this still remains an issue in a minority of the papers included we did highlight this in our limitations section. The role of genetics and epigenetics is also a valid point - however currently only one paper addresses this issue and we had included this in our review. In any case, we have also included this in the limitations section.
3. Further, what happens when someone takes a Super B Complex vitamin? This is not addressed in the manuscript. Another crucial factor is when the patient is diagnosed with PN, is it at that time-point of diagnosis that they are determined to have altered B6 levels, or did the B6 deficiency or excess occur prior to diagnosis. This is not clear.
Thank you for this point. No studies meeting our inclusion criteria used the "Super B Complex" and this is why we did not comment on this. This review, as clearly is stated in our introduction, aims to provide a clearer understanding of B6 related neuropathy (due to excess or due to low levels of B6 and not the role of B6 in the management of neuropathies). However a multivitamin tablet (B-complex) would have included B1 and B12 which also play a role on the management of neuropathy and therefore we wouldn't be able to comment only on the role of B6 which was our aim. Regarding the second point, in the papers where higher B6 levels were linked with neuropathy the period of B6 administration was added in the text and the table, when available. In the papers were low B6 was linked to neuropathy there were no historical data about the period of low levels before the diagnosis of neuropathy - which is a limitation of all nutritional studies that we also included in our limitations section.
4. Table 1 should include another 2 columns with sex of the patients and ethnicity/region they were from. Additionally, a summary chart/bar graph of the demographics obtained should be presented to help visually summarize the key/main point of the manuscript.
We included information about sex as recommended (where this info was available) as well as the region(s) of each study. We did not provide a summary of the demographics as a figure as this in our opinion wouldn't add much - as this is a review of B6 related PN the studies are very different in their methodology and population used.
5. Additionally, the references seem too little for such a review in which 70-80 papers would be more appropriate.
The relatively small number of eligible to be included papers (n=20) reflects that the topic is relatively understudied. We set very strict criteria as we needed only to use appropriate papers. In the PRISMA chart we gave precise information about the reasons for exclusion.
Round 2
Reviewer 3 Report
Accept as is since they have addressed all reviewer comments with qualification/justification.
Accept as is since they have addressed all reviewer comments with qualification/justification.